# Self-consistent kinetic model of nested electron- and ion-scale magnetic cavities in space plasmas

Jing-Huan Li [1,11], Fan Yang [1,11], Xu-Zhi Zhou [1✉], Qiu-Gang Zong [1✉], Anton V. Artemyev[2,3], Robert Rankin[4], Quanqi Shi[5], Shutao Yao [5], Han Liu [1], Jiansen He [1], Zuyin Pu[1], Chijie Xiao[6], Ji Liu[7], Craig Pollock[8], Guan Le[9] & James L. Burch [10]

NASA's Magnetospheric Multi-Scale (MMS) mission is designed to explore the proton- and electron-gyroscale kinetics of plasma turbulence where the bulk of particle acceleration and heating takes place. Understanding the nature of cross-scale structures ubiquitous as magnetic cavities is important to assess the energy partition, cascade and conversion in the plasma universe. Here, we present theoretical insight into magnetic cavities by deriving a self-consistent, kinetic theory of these coherent structures. By taking advantage of the multipoint measurements from the MMS constellation, we demonstrate that our kinetic model can utilize magnetic cavity observations by one MMS spacecraft to predict measurements from a second/third spacecraft. The methodology of "observe and predict" validates the theory we have derived, and confirms that nested magnetic cavities are self-organized plasma structures supported by trapped proton and electron populations in analogous to the classical theta-pinches in laboratory plasmas.

[1] School of Earth and Space Sciences, Peking University, 100871 Beijing, China. [2] Institute of Geophysics and Planetary Physics, University of California, Los Angeles, CA 90095, USA. [3] Space Research Institute, Russian Academy of Sciences, Moscow 117997, Russia. [4] Department of Physics, University of Alberta, Edmonton, AB T6G2G7, Canada. [5] Institute of Space Sciences, Shandong University, Weihai 264209, China. [6] School of Physics, Peking University, Beijing 100871, China. [7] National Space Science Center, Chinese Academy of Sciences, Beijing 100190, China. [8] Denali Scientific, Fairbanks, AK 99709, USA. [9] NASA Goddard Space Flight Center, Greenbelt, MD 20771, USA. [10] Southwest Research Institute, San Antonio, TX 78238, USA. [11] These authors contributed equally: Jing-Huan Li, Fan Yang. ✉email: xzzhou@pku.edu.cn; qgzong@pku.edu.cn

In the plasma universe, unstable velocity distributions of ions and electrons are formed around a variety of acceleration regions such as shock waves and magnetic reconnection sites[1–4], which can excite various types of plasma waves, either compressional or incompressional. The incompressional waves are usually damped very slowly due to the lack of Landau resonance with the ambient plasmas[5], which enables efficient transport of their energy to large distances from the origin. Therefore, it is the incompressional waves that largely determine the nonlinear cascade of large-scale fluctuations towards smaller scales to shape the turbulent electromagnetic spectra[6,7]. The compressional waves, on the other hand, are often believed to contribute only a small fraction of the plasma turbulence (e.g., 5–20% in the solar wind[7,8]). However, even the compressional waves can survive for a long time if their growth is sufficiently strong to reach a nonlinear stage. A classical example of these nonlinear structures is the magnetic cavities, also referred to as magnetic holes, that have been reported to travel in the solar wind for at least several astronomical units[9].

Such magnetic cavities, characterized by quasi-symmetric depressions of magnetic field strength accompanied by plasma density and pressure enhancements, have been intensively investigated in the observations of space and astrophysical plasma environments, including the solar wind[9–12], heliosheath[13], terrestrial magnetosheath[14–16], magnetotail[17–24], planetary[25,26], and even cometary environments[27,28]. These coherent structures have long been found to appear intermittently with size varying from fluid down to ion scales[29,30]. More recently, the availability of the high-resolution observations from the MMS constellation[31] further enables identification of electron-scale kinetic cavities[32], which in specific occasions are found to be embedded within proton-scale cavities[16]. Therefore, these magnetic cavities have been widely believed to play important roles in the energy cascade, conversion, and dissipation in turbulent plasmas[33,34].

Several mechanisms have been proposed to explain the generation of magnetic cavities in space plasmas. A leading candidate is the mirror[9,11,35] or for electron-scale cavities, the electron-mirror instabilities[36–38]. This scenario is supported by observations of anisotropic particle distributions within the cavity, with plasma pressure in the directions perpendicular to the magnetic field often higher than in the parallel direction[11,29,32]. Alternative candidates include nonlinear Alfvén waves and magnetosonic solitary waves[39–42], and it remains unclear as to which mechanism dominates. After their generation, the existence of stable magnetic cavities requires a fine balance between electromagnetic and plasma stresses, which may not be easy to achieve in the electron scale given the decoupled ion and electron motions. Can these electron-scale cavities be stable and quasi-stationary, or are we simply observing some non-steady magnetic field variations?

To better answer these questions, a series of kinetic simulations have been carried out[43,44]. They show that coherent electron-scale cavities, stationary in the plasma rest frame and stable for hundreds of electron gyroperiods, can emerge self-consistently in the course of turbulent plasma evolution. These simulated cavities have circular cross-sections, with azimuthal currents near their edges contributing to the field depression. Given that the ring-shaped, diamagnetic currents are carried solely by electrons in the simulation, these electron-scale magnetic cavities are also referred to as "electron vortex" cavities[34,43].

Despite the success in simulating the cavity emergence, the simulations can only be qualitatively compared to in situ observations[45], and it is important to extract the missing information hidden in the observational data. This is not straightforward; however, since in situ measurements can only achieve the profiles along spacecraft trajectories. The deadlock to resolve the two- or three-dimensional configurations can only be solved by

comprehensive models of magnetic cavities. According to a recently proposed kinetic model[46], magnetic cavities are analogous to classical theta-pinches[47] in laboratory plasmas, with their electromagnetic and particle profiles determined by Vlasov–Maxwell equations in cylindrical coordinates. Many model features, such as the nested structures, ring-shaped current, and enhanced plasma pressure, are all present in spacecraft observations. However, an important limitation is that the modeled particle distributions are shifted-Maxwellians (superimposed over the background, isotropic population), which differs dramatically from the observed anisotropic electron distributions (typically with higher fluxes perpendicular to the magnetic field, see Fig. 1h, i for sample observations). Therefore, it is important to construct a more comprehensive model capable of describing the observational characteristics, first of all the electron kinetic distributions within the magnetic cavities.

Here, we demonstrate existence of such a kinetic equilibrium, and thus prove the survival of electron- and proton-scale, nested magnetic cavities over a macrotime scale. This equilibrium model, constructed by taking into account the adiabatic electron behavior, is used to reconstruct nested cavities observed in ref. [16] by the four-spacecraft MMS constellation[31]. With model parameters derived from single-spacecraft measurements, the reconstructed profiles agree remarkably with observations from all observing spacecraft. Based on our validated model, we then discuss cross-scale properties of the nested cavities.

## Results

**Model development**. Our model is developed in cylindrical coordinates ($\rho$, $\varphi$, $z$), which is adapted to conform to the circular cross-sections of magnetic cavities in both simulations[43] and observations[16,20]. In deriving a kinetic equilibrium state consistent with observations, proton and electron distributions must be taken as functions of the invariants of particle motion[46,48,49] so as to satisfy the Vlasov equation. The invariants of motion used in ref. [46] are the particle energy and the canonical angular momentum in the azimuthal direction. To account for the electron perpendicular anisotropy, we adopt another invariant of electron motion, the magnetic moment $\mu$ (which depends on perpendicular but not parallel velocity). The three invariants are then combined to construct the particle distribution functions, which comprise a current-carrying and a background population for each species to accommodate the multi-scale geometry and the location-dependent anisotropic features. The modeled distribution functions are then integrated to derive particle and current densities, which, when substituted into Maxwell's equations, provide a self-consistent determination of the cross-scale cavity profiles. Note that the model, given appropriate parameters, can also be used to produce single-scale (either electron- or proton-scale) magnetic cavities. More details about the model appear in the "Methods" section.

**Observations**. Our kinetic model detailed in the "Methods" section is used to reconstruct profiles of nested magnetic cavities observed by the MMS constellation on 23 October 2015. During this time interval, the MMS spacecraft were separated from one another by approximately 10 km. The MMS data utilized in this study include the three-dimensional electron and proton velocity distributions (with high time resolution at 30 and 150 ms, respectively) from the fast plasma investigation (FPI) instrument[50], and the electromagnetic field measurements from the FIELDS instrument suite[51]. A small-scale cavity, observed by three MMS spacecraft (MMS1, MMS3, MMS4, with the neighboring MMS2 observing no perturbations that confirms the very small size of the cavity), has been reported as the first

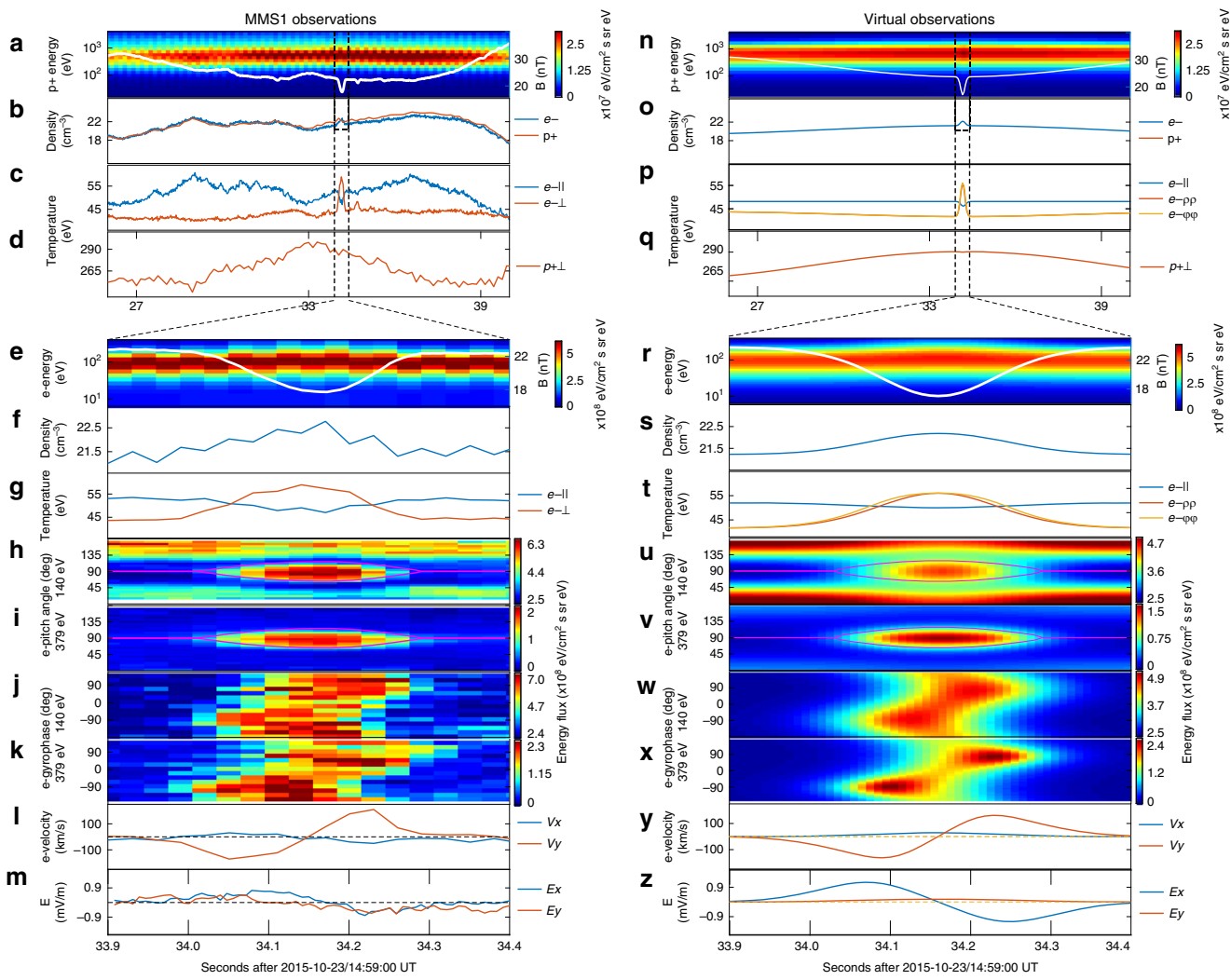

**Fig. 1 Comparison between the MMS1 observations of nested magnetic cavities on 23 October 2015 and the virtual spacecraft observations across the equilibrium model.** Panels **a–d** show the MMS1 observations between 14:59:26UT and 14:59:40UT of the magnetic field strength (white line, superposed over the proton energy spectrogram), the plasma density, the electron parallel and perpendicular temperatures, and the proton temperature, respectively. Panels **e–m** are the zoomed-in view of the electron-scale magnetic cavity within 0.5 s, which show the magnetic field strength (overplotted on the electron energy spectrogram), the electron number density, electron parallel and perpendicular temperatures, the electron pitch angle distributions in the 140- and 379-eV energy channels, the electron gyrophase spectrograms in the same channels, the electron bulk velocity in the rest frame of the proton fluid, and the electric field (Lorentz transformed into the proton rest frame), respectively. Panels **n–z** are the virtual spacecraft observations across the modeled magnetic cavities in the same format as in panels **a–m**. The magenta lines in panels **h** and **i** are the "local loss cones", defined in ref. [16] to represent the trapping regimes between two hypothetical mirror points (with the magnetic field strength of 22.3 nT, or the ambient field strength) bracketing the cavity in the field-aligned direction.

identification of kinetic-scale magnetic cavities in the magnetosheath[15]. The cavity radius, according to an innovative particle sounding technique[16], is about 10 km (a few times greater than electron thermal gyroradius). It was also found in ref. [16] that the electron-scale cavity was embedded within a proton-scale cavity (with a radius of over 500 km).

Figure 1 provides an MMS1 observational overview of nested cavities. Figure 1a–d shows the proton-scale cavity characterized by depressed magnetic field strength (from 32 to 24 nT), enhanced proton density and temperature, and enhanced energy fluxes of protons between 100 and 1000 eV. The electron-scale magnetic cavity appears in the center of the proton-scale cavity, with its 500-ms, zoomed-in view shown in Fig. 1e–m. The characteristic signatures of the cavity, including the depressed magnetic field strength (from 24 to 17 nT) and the enhanced

electron density and temperature, are very similar to the proton-scale cavity, although the profiles are more symmetric than the proton-scale cavity.

To better visualize the different behavior of electrons and protons, the electron data in Fig. 1 are organized in the rest frame of the proton fluid. We also define a coordinate system on the basis of magnetic field and proton bulk velocity measurements (both of which hardly changed direction within the electron-scale cavity), in which the z-axis is the average direction of the magnetic field, the y-axis is along the cross product of the average plasma bulk velocity and the z-direction, and the x-axis completes the triad. The coordinate system, illustrated in Fig. 2 (together with the modeled proton- and electron-scale cavities), is used throughout this paper unless otherwise specified.

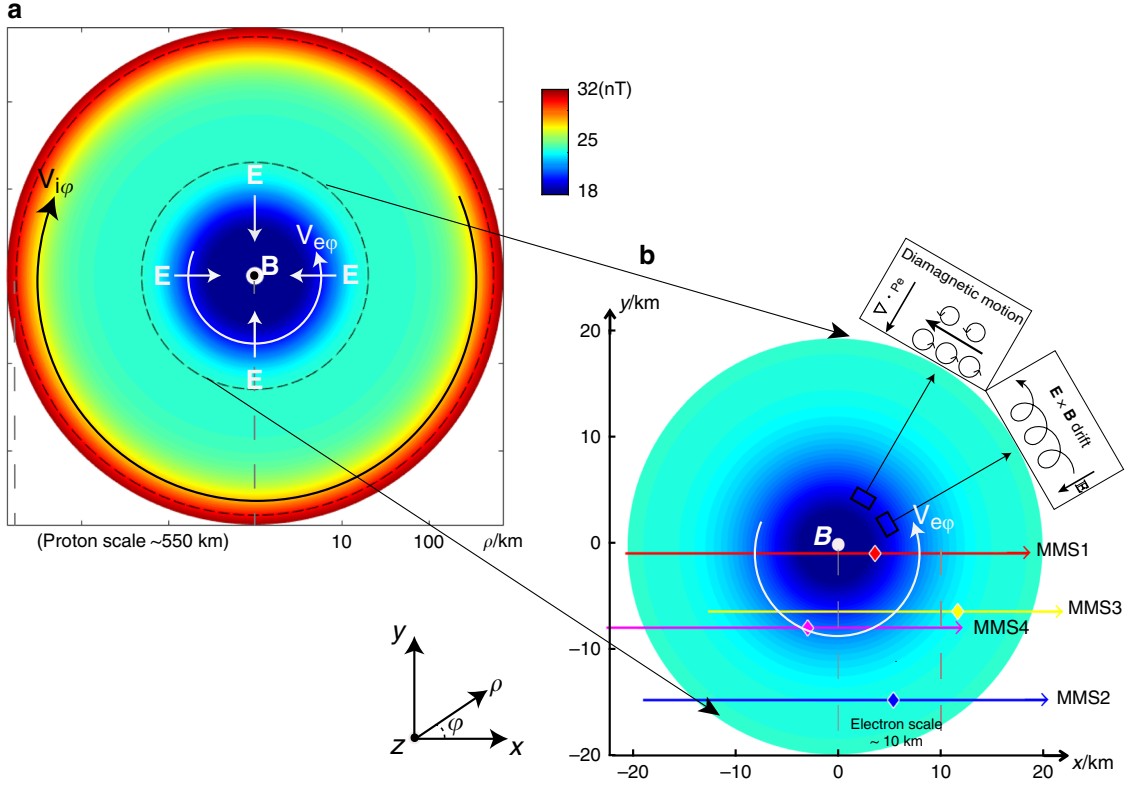

**Fig. 2 Cross-section of the nested magnetic cavities in the perpendicular plane.** Panel **a** shows the nested cavities in the proton scale (~550 km, in logarithmic form), and Panel **b** shows the zoomed-in view within the electron scale (~10 km, in linear form). The color denotes the magnetic field strength profiles. The four lines are the trajectories of the MMS constellation across the nested cavities, with the diamond symbols representing their positions within the modeled cavities at 14:59:34.21 UT. The definition of the Cartesian ($x$, $y$, $z$) and the cylindrical ($\rho$, $\varphi$, $z$) coordinates are given in the bottom-left corner. The directions of the electric field, proton and electron bulk velocities are also indicated as arrows in the plots. The two insets in panel **b** illustrate the nature of the electron diamagnetic motion and their $\mathbf{E} \times \mathbf{B}$ drift, respectively.

Figure 1h, i presents the pitch angle spectra of electron fluxes in the 140- and 379-eV channels. Within the electron-scale cavity, the electron fluxes in both channels are significantly enhanced in the direction perpendicular to the magnetic field, although the field-aligned electron fluxes are higher outside the electron-scale cavity (also see Fig. 1g for the perpendicular and parallel temperature variations). The strong anisotropy within the electron-scale cavity has been attributed to two magnetic mirrors bracketing the cavity in the field-aligned direction[16], between which electrons with near-90° pitch angles (outside the "local loss cones") are trapped by magnetic mirror force. This is not the only interpretation though; we will show that the observed pitch angle spectrum can be reproduced in our cylindrical model without invoking adjacent magnetic mirrors.

Another important feature for this event is the gyrophase dependence of the electron fluxes, even in the plane perpendicular to the magnetic field. The apparent non-gyrotropic distributions for the 140- and 379-eV electrons are shown in Fig. 1j, k, in which 0° and 90° represent the electron motion in the $+x$ and $+y$ directions, respectively. This non-gyrotropic feature is most significant near the edges of the electron-scale cavity, with higher fluxes at −90° and 90° gyrophase in the leading and trailing edges, respectively. The gyrophase dependence also corresponds to a bipolar profile of electron bulk velocity in the $xy$ plane (see Fig. 1l), which are predominantly in the $-y$ and $+y$ directions in the leading and trailing edges, respectively. Note that the observed electric field, after being Lorentz transformed into the rest frame of the proton fluid, also shows bipolar signatures in the $x$ direction (see Fig. 1m).

**Modeling results**. We reconstruct the nested cavities based on our equilibrium model, and simulate observations a virtual spacecraft would make when following the MMS1 orbit across the cavities in the $+x$ direction, which can be equivalently understood as magnetic cavities moving at the proton flow speed to encounter the immobile spacecraft. The reconstruction and simulation procedures are described in the "Methods" section. Virtual spacecraft observations are shown in Fig. 1n–z, which have one-to-one correspondence to the MMS1 observations in Fig. 1a–m. The virtual observations display all the aforementioned characteristics, including multi-scale magnetic depressions and density enhancements, electron anisotropy and non-gyrotropy, and bipolar electron velocity and electric field variations in the electron-scale cavity.

MMS3 and MMS4 observations are displayed in Figs. 3a–i and 4a–i, respectively. Their direct comparison with the model (Figs. 3j–r and 4j–r), for two virtual spacecraft that follow the MMS3 and MMS4 orbits, shows excellent agreement across the electron-scale magnetic cavity. Especially, we note that the electron gyrophase spectrogram, the electron bulk velocity, and the electric field observations all show more complicated signatures than those in MMS1 data, but they can still be reproduced without changing any model parameters. The agreement, therefore, provides confidence on the model validity.

**Discussion**
The validated model provides detailed electromagnetic and particle profiles of the nested magnetic cavities that enable us to better

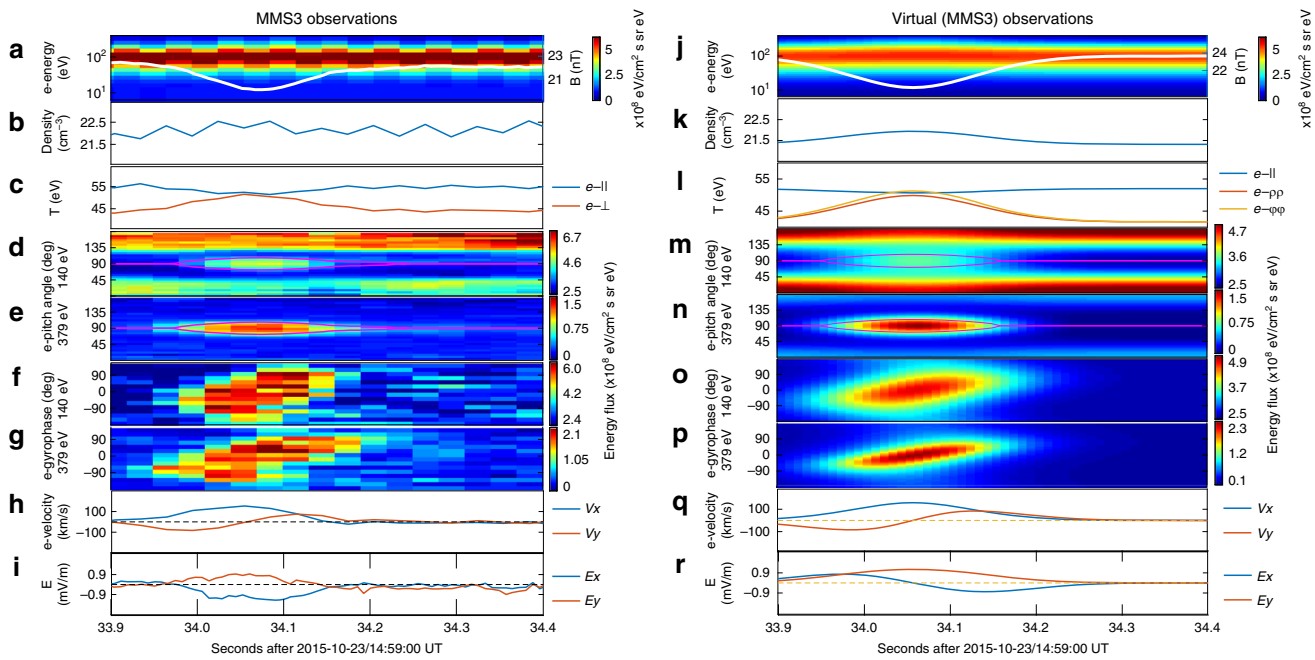

**Fig. 3 The MMS3 observations of the electron-scale magnetic cavity in comparison with virtual spacecraft observations of the modeled cavity.** Panels **a**–**i** and **j**–**r**, both in the same format as in Fig. 1e–m, are the MMS3 and the virtual spacecraft observations, respectively.

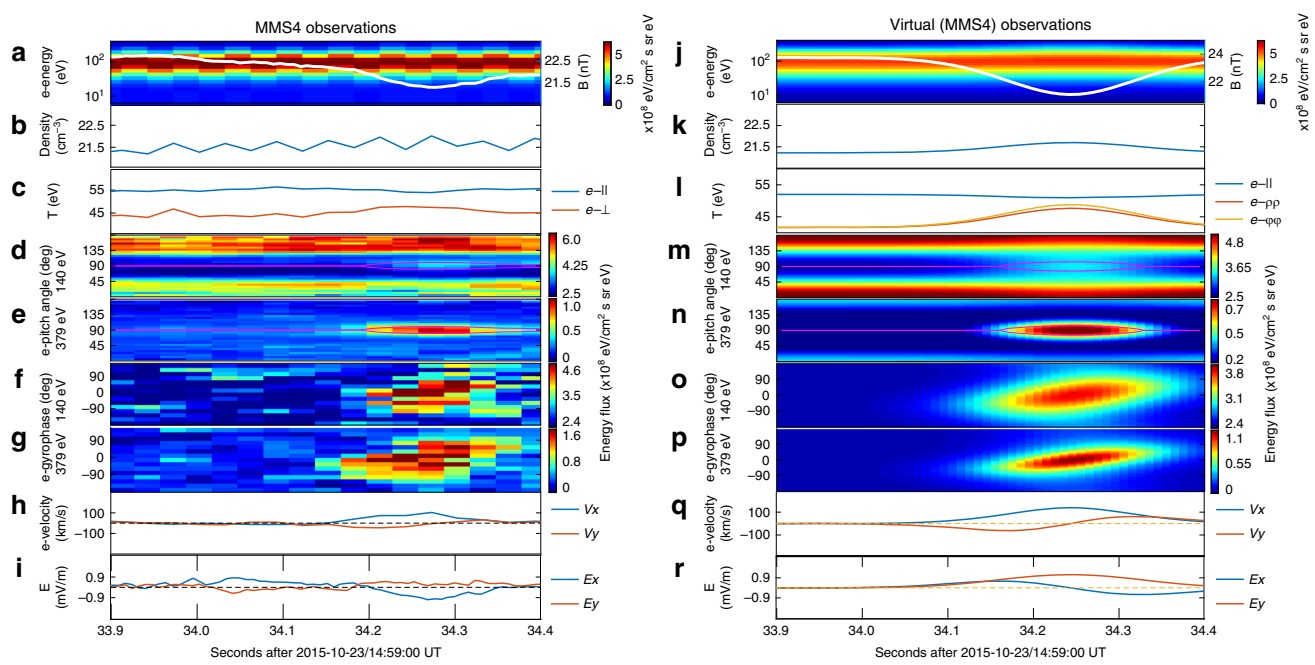

**Fig. 4 The MMS4 observations and virtual spacecraft observations of the electron-scale magnetic cavity.** Panels **a**–**r** are in the same format as in Fig. 3.

understand the roles of electrons and protons in the formation of cross-scale structures. These profiles in cylindrical coordinates are shown in Fig. 5, with the horizontal axis being the radial distance from the cavity center (in logarithmic form for a better visualization of the cross-scale geometry, also given in units of proton and electron thermal gyroradii). It is evident that the magnetic strength depression and plasma density enhancement both show multi-scale features (also see Fig. 2 for schematic illustration of multi-scale magnetic strength profiles), whereas the variations of proton and electron temperatures appear only on proton and electron gyro-scales, respectively. Given the anti-correlation between magnetic

field strength and plasma density/temperature, one may speculate that the multi-scale magnetic depressions are caused by the proton and electron diamagnetic currents in the azimuthal direction. Such an azimuthal flow will manifest in Cartesian coordinates, for any spacecraft moving across the magnetic cavity, as bipolar velocity variations in the $y$ direction (see Figs. 1l, 3h, and 4h). For this specific event, however, it was found in ref. [16] that the electron diamagnetic velocity (computed from the inferred electron pressure profile) is lower than the observed electron bulk velocity (see Fig. 5k). This discrepancy can be examined on the basis of our kinetic model.

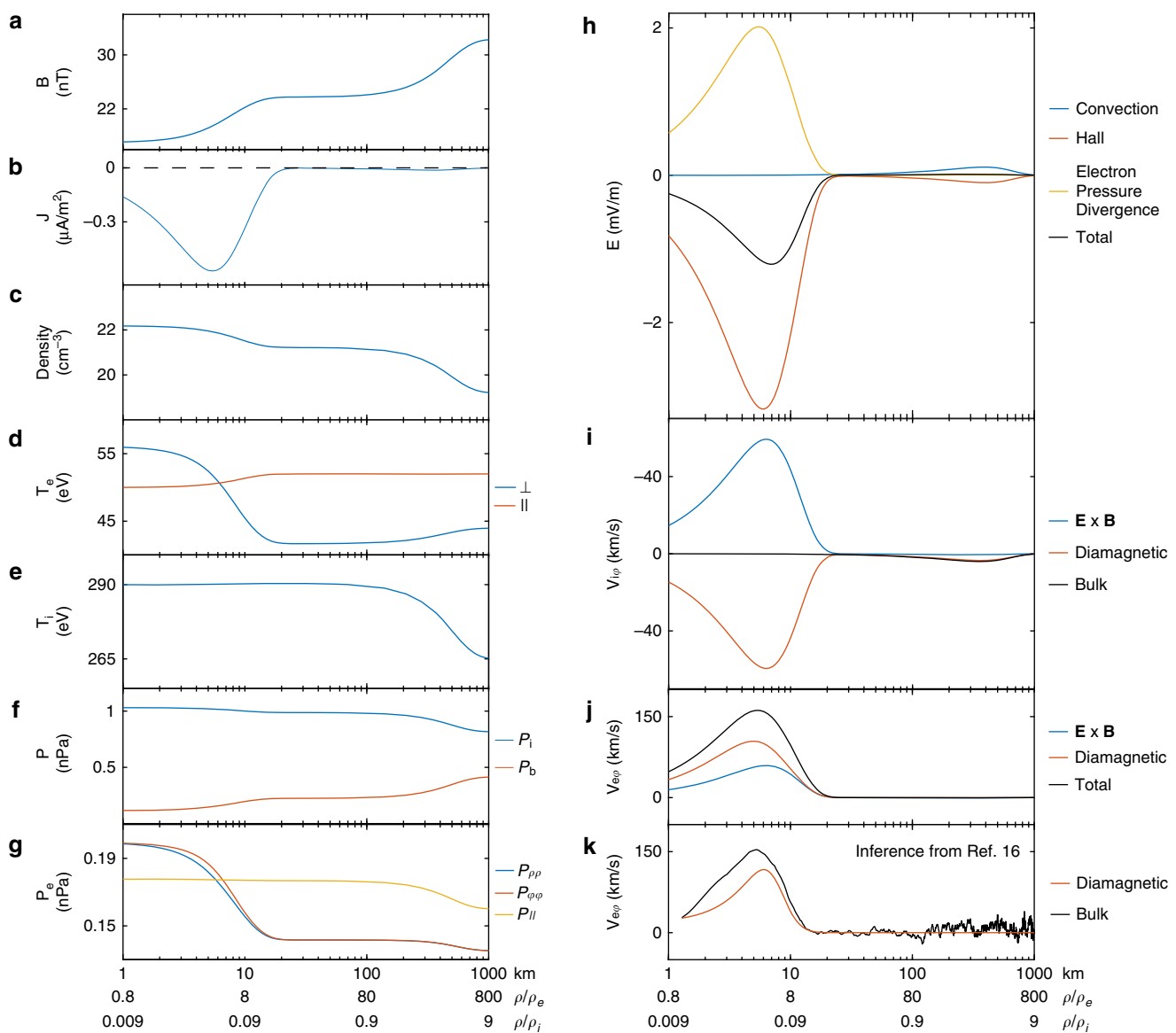

**Fig. 5 The modeled electromagnetic field and particle profiles of the nested cavities.** The horizontal axis represents the distance from the cavity center in the logarithmic scale, also normalized by thermal electron and proton gyroradii. Panels **a**–**g** show the radial profiles of the magnetic field, electric current, plasma density, electron parallel and perpendicular temperatures, proton temperature, proton and magnetic pressure, and the diagonal terms of the electron pressure tensor. Panel **h** shows the distributions of the radial electric field, which according to the generalized Ohm's law can be divided into the convection, Hall, and electron pressure divergence terms. Panels **i** and **j** show the proton and electron bulk velocities in the azimuthal direction (together with their **E × B** and diamagnetic components), respectively. Panel **k** shows the electron bulk velocity inferred from ref. [16] in direct comparison with panel **j**.

In the model, the azimuthal electron velocity stems from the dependence of electron phase space density on canonical angular momentum in the azimuthal direction (see Eq. (10) in the "Methods" section, which applies for the current-carrying component of the electron population). At any given location, the canonical angular momentum (and, therefore, the phase space density of the current-carrying electron component) depends on the particle azimuthal speed, which contributes to the non-gyrotropic distributions observed in any energy channel (such as those shown in Figs. 1j, 3f and 4f). The azimuthal flow (or equivalently, the so-called electron vortex) is merely the macroscopic manifestation of the form of the electron distributions. In addition, the coexistence of two electron components with different angular bulk velocities (the background and the current-carrying components, see Eqs. (8) and (10) in the "Methods" section) also contributes to the observed electron non-gyrotropy.

One may also recall the equilibrium momentum equation of the electron fluid,

$$nm_e(\mathbf{u}_e \cdot \nabla)\mathbf{u}_e = ne(\mathbf{E} + \mathbf{u}_e \times \mathbf{B}) - \nabla \cdot \mathbf{P}_e, \qquad (1)$$

which indicates a balance between the centrifugal force of the electron fluid (approximately two orders of magnitude weaker than the other two forces, and hence neglected hereinafter), the electromagnetic force and the divergence of the electron pressure tensor. Here the pressure tensor is contributed only by diagonal terms, with three scalar pressures, $P_{e,\rho\rho}$, $P_{e,\varphi\varphi}$ and $P_{e,//}$, in the radial, azimuthal, and magnetic field directions, respectively. One may find in Fig. 5g that there is a minor difference between $P_{e,\rho\rho}$ and $P_{e,\varphi\varphi}$, which also originates from non-gyrotropic electron distributions (see refs. [52,53] for similar effects associated with non-Gaussian distributions in the reconnection region and in the turbulent magnetosheath, respectively). The divergence of the

pressure tensor can thus be calculated, to be only in the radial direction:

$$\nabla \cdot \mathbf{P}_e = \left[ \frac{\partial P_{e,\rho\rho}}{\partial \rho} + \frac{1}{\rho} \left( P_{e,\rho\rho} - P_{e,\varphi\varphi} \right) \right] \mathbf{e}_\rho. \qquad (2)$$

We next take the crossproduct of Eq. (1) with $\mathbf{B}$ to have

$$\mathbf{u}_{e\perp} = \frac{\mathbf{E} \times \mathbf{B}}{B^2} + \frac{\nabla \cdot \mathbf{P}_e \times \mathbf{B}}{neB^2}, \qquad (3)$$

which indicates that the $\mathbf{E} \times \mathbf{B}$ drift and the diamagnetic motion both contribute to the electron bulk velocity perpendicular to the magnetic field (see Fig. 2b for a schematic sketch). Figure 5j shows these two velocities (together with their sum) in the model, which supports the observation that diamagnetic motion only contributes ~70% of the electron azimuthal bulk velocity[16]. The remaining portion of the azimuthal flow is contributed by the $\mathbf{E} \times \mathbf{B}$ drift in association with the radially inward electric field, which resolves the apparent discrepancy in ref. [16].

We also examine the profiles of the proton bulk velocity and its contributing components (the $\mathbf{E} \times \mathbf{B}$ drift and the proton diamagnetic motion, see Fig. 5i). These two components cancel out in the electron-scale cavity, whereas at the proton scale, the azimuthal flow is dominated by the diamagnetic motion (albeit much weaker than in the electron scale). These profiles also enable us to quantify the current carriers in nested cavities. At the proton scale, the diminishing electron flow (Fig. 5j) and the absence of the electric field indicate that the proton diamagnetic motion is solely responsible for carrying the azimuthal current. At the electron scale, the negligible proton flow (Fig. 5i) suggests that the azimuthal current is carried only by electrons, with their $\mathbf{E} \times \mathbf{B}$ and diamagnetic motion both contributing a significant portion. This is consistent with the observations in ref. [22], although in other studies both terms have been proposed as the sole contributor[15,21]. The very different electron and proton flow velocities, shown in Fig. 5i, j, are also responsible to the dramatic difference between the current intensities in the electron and proton scales (see Fig. 5b). We should point out here that within the electron-scale cavity, the fine balance between the proton diamagnetic motion and the $\mathbf{E} \times \mathbf{B}$ drift indicates that the current density profile can be alternatively understood to be determined by the sum of the electron and proton pressure divergences. In other words, the protons still play a role in regulating the electron-scale cavity profiles (via polarization electric field) despite their negligible flow speed within this scale.

The above discussion implies the importance of understanding the origin of the electric field in the radially inward direction (see Fig. 5h), which is strongest near the edge of the electron-scale cavity. According to the generalized Ohm's law,

$$\mathbf{E} = -\mathbf{u} \times \mathbf{B} + \frac{\mathbf{j} \times \mathbf{B}}{ne} - \frac{\nabla \cdot \mathbf{P}_e}{ne} - \frac{m_e}{e}(\mathbf{u}_e \cdot \nabla)\mathbf{u}_e, \qquad (4)$$

the electric field in an equilibrium plasma system is balanced by the four terms on the right-hand-side of Eq. (4), referred to as the convection, the Hall, the electron pressure divergence, and the inertial terms, respectively. Here the inertial term corresponds to the very weak centrifugal force and can be safely neglected. The first two terms on the right-hand side may also be combined as $-\mathbf{u}_e \times \mathbf{B}$, which represents the motional electric field of the electron fluid (and turns Eq. (4) into Eq. (1)). The three right-hand-side terms in Eq. (4) are shown in Fig. 5h. At the proton scale, the electron pressure divergence term is negligible (indicating the frozen-in of the electron fluid to the proton-scale cavity), and the convection and Hall terms are balanced to produce a negligible electric field. The frozen-in condition for the proton fluid, on the other hand, is not satisfied ($\mathbf{E} \neq -\mathbf{u} \times \mathbf{B}$). At

the electron scale, the convection term becomes negligible, whereas the radially inward Hall term (resulting from the decoupled proton and electron fluid motion, or equivalently, the nonzero electric current) and the radially outward electron pressure divergence term both contribute to the electric field (the former being stronger to result in an inward net electric field). Similar observations have been also made in other small-scale structures, most noteworthy the electron diffusion region of magnetic reconnection, in which the electron pressure divergence term becomes very important only next to the Hall term[54] or even dominant[55,56] in the generalized Ohm's law.

We note that the modeled profiles are $z$-independent, which indicates the absence of magnetic mirror structures. In other words, the enhanced electron anisotropy (with higher fluxes near-90° pitch angles, see Figs. 1i, 3e and 4e) in the cavity center may not necessarily be associated with the conventional interpretation[16] of trapped electrons between adjacent mirrors. In our model, the enhanced electron anisotropy stems from the $\mu$-dependence of the electron phase space densities. At any given electron energy, the magnetic moment becomes larger at pitch angles closer to 90° and/or at locations with lower magnetic field strengths. The perpendicular and parallel anisotropy inside and outside the electron-scale cavity (most clear in Fig. 1g, h) corresponds to the positive and negative $\mu$-dependence of phase space densities for the current-carrying and background electron populations (arising from the model parameters $b_{e,1} < 0$ and $b_{e,0} < 0$ in Eqs. (10) and (8) in the "Methods" section), respectively. This picture may also be understood in the framework of pressure balance in the plane perpendicular to the magnetic field. In the cavity center, the depressed magnetic pressure has to be compensated by the enhanced electron and proton thermal pressure (see Fig. 5f). The electron parallel pressure, on the other hand, changes very slightly in the cavity center. It is this different trend that corresponds to the enhanced perpendicular anisotropy in the cavity center.

We should also point out that despite the profound similarity between the modeled and the observed signatures, there remains a minor difference in that the MMS1 observations of the electric field $E_y$ (in the azimuthal direction, see Fig. 1m for its bipolar variations) cannot be reproduced since the modeled electric field is radially inward. The observed $E_y$, albeit weak in magnitude, is in the same direction as the azimuthal electric current, and therefore indicates the conversion of electromagnetic energy to plasma thermal and/or kinetic energy. A possible mechanism to generate the azimuthal electric field is electromagnetic induction associated with cavity shrinkage, a process reported recently via MMS observations[57–59]. This could indeed happen in this event, since the observed magnetic field (in Fig. 1e) variations are slightly asymmetric with sharper gradient in the trailing edge. These features are certainly beyond the scope of our equilibrium model; however, our model may provide an initial condition for understanding the cavity kinetic evolution and associated particle energization processes[59].

In summary, we are inspired by the similarity between theta-pinches and magnetic cavities to develop a kinetic, equilibrium model of cross-scale magnetic cavities in the space environment. The model shows excellent agreement with spacecraft observations, which indicates the surprising formation of quasi-equilibrium magnetic cavities during the turbulent evolution of space plasmas. The existence of long-lived, electron-scale cavities may be associated with the parallel electron anisotropy within a proton-scale cavity (see Figs. 1g and 5d), which provides a background plasma environment favorable for the maintenance of such coherent, electron-scale structures[43]. These magnetic cavities, traveling with the plasma flows over a macrotime scale, can transport the hot trapped electrons away from the energy

release source and shape the spectrum of compressional fluctuations on scales as small as the electron thermal gyroradius. In fact, the reported magnetic cavities were observed in the terrestrial magnetosheath, a location where compressional turbulence has been proposed and observed to be more important than in the solar wind plasmas[60–62]. Given the complicated dynamics of energy cascade and dissipation[63,64] indicated by the presence of distinct breakpoints in the turbulent magnetic spectrum at electron gyroscales[65,66], the analyzed magnetic cavities may also be considered intermittent structures produced by the inhomogeneity of electron-scale turbulence cascade. The model/observation comparison shows that such cavities represent long-living magnetic structures responsible for steady spectrum of compressional magnetic field fluctuations at electron scales. The successful application of the model also enables us to evaluate the relative importance of various terms in the generalized Ohm's law for these kinetic-scale structures. Finally, our study also provides a theoretical solution, previously stated as difficult to derive[43], that can be used for analysis of plasma instabilities and waves[67,68] and/or serve as initial conditions for kinetic simulations of cavity evolution.

## Methods

Our cavity model provides an equilibrium solution to the Vlasov-Maxwell equations in the cylindrical coordinates ($\rho$, $\varphi$, $z$). In this model, the magnetic vector potential only has azimuthal component, i.e., $\mathbf{A} = A(\rho)\mathbf{e}_\varphi$, which indicates that the magnetic field is always in the $z$ direction. The electric scalar potential $\phi$ also depends on $\rho$, indicating that the electric field is in the radial direction. To construct the proton and electron distribution functions, three invariants of motion are utilized, which include the particle energy, the canonical angular momentum in the azimuthal direction, and the magnetic moment (the so-called first adiabatic invariant, applicable to electrons but not protons in our model given the weak magnetic variations within an electron thermal gyroradius):

$$H_\alpha = \frac{M_\alpha \mathbf{v}^2}{2} + q_\alpha \phi, \quad (5)$$

$$P_{\varphi\alpha} = \rho\left(M_\alpha v_\varphi + q_\alpha A\right), \quad (6)$$

$$\mu_e = \frac{M_e|\mathbf{v}_\perp - \mathbf{v}_D|^2}{2B}, \quad (7)$$

where $\alpha = $ i, e stands for protons and electrons, respectively, and $\mathbf{v}_D$ denotes the electron guiding center drift velocity, which consists of the $\mathbf{E} \times \mathbf{B}$ and the magnetic gradient drifts. We next follow ref. [46] to assume that each species have two distinct components, a background population and a current-carrying population (represented by subscripts 0 and 1, respectively), so the particle distribution functions can be given by

$$f_{e,0} = \delta N_e \left(\frac{M_e}{2\pi\theta_{e,0}}\right)^{3/2} \exp\left(-\frac{H_e + b_{e,0}\mu_e}{\theta_{e,0}}\right), \quad (8)$$

$$f_{i,0} = \delta N_i \left(\frac{M_i}{2\pi\theta_{i,0}}\right)^{3/2} \exp\left(-\frac{H_i}{\theta_{i,0}}\right), \quad (9)$$

$$f_{e,1} = (1-\delta)N_e \left(\frac{M_e}{2\pi\theta_{e,1}}\right)^{3/2} \exp\left(-\frac{H_e - \Omega_e P_{\varphi e} + b_{e,1}\mu_e}{\theta_{e,1}}\right), \quad (10)$$

$$f_{i,1} = (1-\delta)N_i \left(\frac{M_i}{2\pi\theta_{i,1}}\right)^{3/2} \exp\left(-\frac{H_i - \Omega_i P_{\varphi i}}{\theta_{i,1}}\right), \quad (11)$$

where $\delta \in [0,1]$ regulates the densities of the two populations; $b_{e,0}$ and $b_{e,1}$ are constants with the dimension of magnetic field, which provides the perpendicular anisotropy for the background and current-carrying electron population; $N_\alpha$, $\theta_\alpha$ and $\Omega_\alpha$ are constants representing the nominal plasma density, temperature, and angular bulk velocity of species $\alpha$, respectively. It is the different $\Omega_i$ and $\Omega_e$ values that provide the electric current and contribute to the magnetic field variations across the cavity. Therefore, one may adjust the $\Omega_\alpha$ parameters, or more precisely the $\theta_{\alpha,1}/\Omega_\alpha$ values, to regulate the spatial scales of the cross-scale (or single-scale, depending on the selected parameters) cavity.

The substitution of Eqs. (5)–(7) into the distribution functions suggests that the two proton populations have Maxwellian and shifted-Maxwellian distributions,

whereas the distributions of the two electron populations are approximately bi-Maxwellian and shifted bi-Maxwellian (since $\mathbf{v}_D$ is much lower than $\mathbf{v}_\perp$ for most electrons). We next compute the number density $n_\alpha$, the bulk velocity $\mathbf{u}_\alpha$, the current density $\mathbf{j}_\alpha$, and the pressure tensor $\mathbf{P}_\alpha$ for each species, by analytically (for protons) or numerically (for electrons) computing the zeroth, first, and second moments of the distribution functions:

$$n_\alpha = \int \left[f_{\alpha,0}(\mathbf{r},\mathbf{v}) + f_{\alpha,1}(\mathbf{r},\mathbf{v})\right]d\mathbf{v}, \quad (12)$$

$$\mathbf{j}_\alpha = \int (f_{\alpha,0}(\mathbf{r},\mathbf{v}) + f_{\alpha,1}(\mathbf{r},\mathbf{v}))\mathbf{v}d\mathbf{v} = n_\alpha q_\alpha \mathbf{u}_\alpha, \quad (13)$$

$$\mathbf{P}_\alpha = M_\alpha \int (f_{\alpha,0}(\mathbf{r},\mathbf{v}) + f_{\alpha,1}(\mathbf{r},\mathbf{v}))(\mathbf{v} - \mathbf{u}_\alpha)(\mathbf{v} - \mathbf{u}_\alpha)d\mathbf{v} = n_\alpha k \mathbf{T}_\alpha, \quad (14)$$

where the current $\mathbf{j}_\alpha$ is in the azimuthal direction, and the electron pressure tensor $\mathbf{P}_e$ has three diagonal terms ($P_{e,rr}$, $P_{e,\varphi\varphi}$, and $P_{e,//}$) due to the $\mu$-dependence of electron phase space density. Based on these moment calculations, we are now able to examine the Ampere's law and the Poisson's equation, although in practice the latter is approximated by the quasi-neutrality condition (since the spatial scale of interest is much larger than the Debye length, and the net charge density is several orders of magnitude smaller than the electron or proton density). In the cylindrical coordinates, we have

$$\frac{\partial}{\partial\rho}\left[\frac{1}{\rho}\frac{\partial}{\partial\rho}(\rho A)\right] = -\mu_0\left(\mathbf{j}_e + \mathbf{j}_i\right), \quad (15)$$

$$n_e = n_i, \quad (16)$$

which can be solved (with Eqs. (12)–(14) substituted) numerically for any given series of parameter sets ($\delta$, $N_e$, $\Omega_e$, $\Omega_i$, $\theta_{e,0}$, $\theta_{e,1}$, $\theta_{i,0}$, $\theta_{i,1}$, $b_{e,0}$, $b_{e,1}$, and the boundary condition $B|_{\rho=0}$). Note that $N_i$ is not an independent parameter due to the quasi-neutrality condition (16). The solution yields the profiles of $A$ and $\phi$ (and therefore the electromagnetic field profiles), which can be substituted into Eqs. (5)–(11) to determine the distributions of the electron and proton phase space densities.

We should note that the electromagnetic field profiles are required in the construction of the invariants of motion (especially the magnetic moment, which depends on the electron drift velocity $\mathbf{v}_D$) before these profiles are self-consistently determined. Therefore, we take advantage of an iteration procedure starting from uniform magnetic and zero electric field profiles, which are substituted into the model for updated profiles until they converge (with the magnetic field difference less than 0.1%) to the final equilibrium solution.

The model parameters are determined from the MMS1 observational data. In this specific event, the determination becomes even more straightforward since the MMS1 orbit intersected the cavity center[16], which suggests that the boundary condition $B|_{\rho=0}$ equals the minimum magnetic field strength observed along the orbit (17 nT). The other parameters are determined on the basis of optimized match with the observed plasma density, electron bulk velocity (transformed into the proton fluid rest frame), perpendicular and parallel temperature, and magnetic field profiles. Note that although the MMS3 and MMS4 observations, together with the MMS1 electric field data, show compelling similarities with the modeling results (see Figs. 1, 3 and 4), they are not used in the parameter determination procedure. The adopted parameters are as follows: $\delta = 0.88$, $N_e = 23.7$ cm$^{-3}$, $\Omega_e = 52.6$ s$^{-1}$, $\Omega_i = -0.158$ s$^{-1}$, $\theta_{e,0} = 52$ eV, $\theta_{e,1} = 45.2$ eV, $\theta_{i,0} = 265$ eV, $\theta_{i,1} = 473.3$ eV, $b_{e,0} = 5.89$ nT, and $b_{e,1} = -10.36$ nT.

Given these model parameters, the electron and proton distribution functions and the self-consistent electromagnetic field profiles are determined after four iterations, which enable us to simulate the virtual spacecraft observations. Here we have four immobile spacecraft positioned with the same configuration and separation as the MMS constellation, and have the modeled nested cavities moving along the $-x$ direction at the speed of 78 km/s (the average proton bulk velocity perpendicular to the magnetic field).

## Data availability

The datasets analyzed in this study are publicly available from the MMS science data center (https://lasp.colorado.edu/mms/sdc/public/), including burst-mode electromagnetic fields from the Fluxgate Magnetometers (fgm/brst/l2/v4.18.1) and the Electric field Double Probe (edp/brst/l2/dce/v2.2.0), together with the proton and electron distributions from the Fast Plasma Investigation (fpi/brst/l2/v3.3.0).

## Code availability

All the relevant codes, including the Matlab routines, the demo inputs, and a readme instruction, are available from Github (https://github.com/lijinghuan1997/cavitymodel) and also from Zenodo[69].

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

## Acknowledgements

We are grateful to the MMS mission and the MMS team for providing high-quality, high-resolution electromagnetic field and plasma measurements. The study carried out in Peking University was supported by National Natural Science Foundation of China (NSFC) grants 41774168 and 41421003. A.V.A. acknowledges the support by Russian Science Foundation (RSF) grant No. 19-12-00313. R.R. acknowledges the support from Canadian Space Agency (CSA) and the Natural Sciences and Engineering Research Council (NSERC).

## Author contributions

J.-H.L. and F.Y. developed the theoretical model, analyzed the observational data, led the model-observation comparison, and co-wrote the paper. X.-Z.Z. and Q.-G.Z. designed the project and oversaw the progress. X.-Z.Z. also took responsibility for the paper writing and revision. A.V.A. contributed to the model development and the paper revision. R.R. contributed to the data interpretation and the paper writing. Q.S., S.Y., H.L., J.H., Z.P., C.X., and J.L. assisted with data analysis and interpretation. C.P., G.L., and J.L.B. assured the quality of the MMS data.

## Competing interests

The authors declare no competing interests.
