## [Peer Review File · Nature Communications]

Report on the paper 242255 submitted by Li et al. to Nature Communications (2020):

Nested electron and proton-scale scale magnetic cavities: Manifestation of kinetic theta-pinch equilibrium in space plasma

The submitted paper is a continuation of the paper published in NCOMMS by Liu et al. (2019), see ref. 7, where some members of this team have analyzed MMS data on kinetic scales of only tens of kilometers. Using the unprecedented very high-resolution of 30 (150) milliseconds for electron (proton) distribution functions, and 8 milliseconds for magnetic field instruments, they have found that the magnetic hole is nearly circular with a radius of about 10 km, i.e. extremely small in the space environment. Now the present authors compare MMS observations with the results of a self-consistent kinetic theory. The paper is also interesting and worth publishing for a general audience, but I am concerned about the following issues

Abstract

the title: magnetic cavities or better magnetic holes (see lines 45, 46)

line: 35 (and 84 – 86) It is not clear from the abstract why it is important to predict measurements from one to a second/third MMS spacecraft. Maybe because of an anomaly on MMS 4 from June/July 2018, e.g., see and add ref. in the text *Astrophys. J. Lett*, 885:L26 (2019).

Model

Please refer to all panels in Fig. 1 in the text (not only in captions) possibly in order

line 113: Why MMS2 observed no perturbations?

Line 120 ff: e.g. discuss the depressed magnetic field in Fig. 1a

line 142 ff: What are the other interpretations?

l. 146 the apparent non-gyrotropic features has been reported also from other spacecraft, e.g. THEMIS, *ApJL*, 851:L42, 2017.

l. 183: show proton and electron gyroscs on the axes and explain and justify the speculation that magnetic depressions are caused by diamagnetic currents.

The discrepancy on the basis of your kinetic model (line 192ff):

line 202: why only diagonal terms are taken in Eq. (1). In the reconnection region (line 207, see e.g., ref. *ApJL* 885:L26, 2019) the electrons are decoupled from ions are not anymore convected by the magnetic field, and the equilibrium fluid equation is not anymore applicable. In the generalized Ohm's law of Eq. (4) an inertial term is missing. It should be justify why this term is not important.

l. 244: At the electron scale, when electrons decouple, the divergence term dominates the electric field and in the case of reconnection it should be even stronger than the Hall (opposite to your case as you say in the parenthesis, see e.g., *GRL* 46, 10295; *ApJL* 885:L26, 2019).

In summary: please say more clearly about any physical interpretation justifying that your equilibrium model agrees with observations. Why a theoretical solution is 'hard to derive'? Discuss possible applications for multi-scales observations in space plasmas.

REVIEWER COMMENTS

Reviewer #2 (Remarks to the Author):

The paper presents observations from the MMS spacecraft of electron and ion scale magnetic cavities to which several dynamical features are associated (e.g., density enhancement, local electron perpendicular heating). The observations are compared to synthetic data obtained using a virtual spacecraft that crosses similar cavities obtained from a theoretical model built by the authors. A very good agreement is found between the model-based data and those of MMS. This type of approach allows one to gain insight into the dynamics of magnetic cavities in space plasmas. The paper is concise and well written, although a bit too technical.

Ion and electron scale magnetic cavities have been previously reported from MMS observations, some of the related papers are cited by the authors. The added value of the present paper is the theoretical model that provides a possible framework to interpret MMS observations of the magnetic cavities. This is an interesting piece of work that helps better understanding the dynamics of this type of coherent structures forming turbulent plasmas. However, the paper appears to this referee as incremental and of a low-to-moderate impact to the current knowledge in the field. As said above, the observations are not new and other theoretical/numerical works have been already proposed (although not necessarily with the same level of detail). It is true that the evidenced structures can locally energize particles, but what is the impact of such energization on the dynamics on larger (macro) scales or on the modelling of general problems of plasma physics is not clear. This contrasts with the claim in the abstract about the potential broad impact of the work (e.g., "Understanding the nature of cross-scale structures ubiquitous as magnetic cavities is important to assess the energy partition, coupling and cascade in the plasma universe"). Also, the multi-scale coupling claimed in different locations of the paper is not strongly supported by the observations: the sole co-existence of two coherent structures with two characteristic scales –proton and electron gyro-radii, can hardly be considered as a multi-scale coupling as known in turbulent or complex systems.

For all these reasons I do not recommend the publication of the paper in Nature Com. I would rather recommend submitting it to a more specialized journal in space or plasma physics (e.g., JGR, JGR, Phys. Plasmas) where similar papers are regularly published (after considering the comments above).

Reviewer #3 (Remarks to the Author):

This manuscript by Li et al. constructs a self-consistent, cylindrically symmetric, equilibrium model for the electromagnetic fields and particle distribution function within nested electron and ion scale magnetic cavities, by considering the conservation of energy, canonical angular momentum, and magnetic moment. The model is applied to observations of nested ion and electron-scale magnetic cavities by the Magnetospheric Multiscale mission, which have been discussed from an observational perspective. The model appears to successfully reproduce many features observed within the observations, not only for the one spacecraft that was used to set the parameters of the model, but also for the other two spacecraft which encountered the structure.

Kinetic scale magnetic cavities have been reported across a number of different space plasma environments, including the solar wind, Earth's magnetosheath, Earth's magnetopause, and Earth's plasma sheet, and more recently nested ion and electron-scale cavities have been reported. Such structures have been suggested to be related to turbulence in these regions and may play some role dissipation, although, to my knowledge, a full understanding of their formation and significance within the plasma has not been fully determined. This paper is significant in that it presents a new self-consistent model for the fields and particle distribution functions.

Overall, I find the paper well written and believe it will likely be suitable for publication with some revision and further discussion. In particular, I think further discussion of the significance and usefulness of the new model would be warranted, particularly for a Nature publication, and I have some questions about the origin of the electric fields in the structure that may prompt further discussion within the paper. I have outlined this and other comments in detail below.

1) Does this model only work for "electron-vortex" magnetic cavities that are embedded within ion-scale cavities? To my knowledge, most of the observed electron-scale magnetic cavities are not reported within a larger ion scale cavity, so I think it is important to discuss whether this model could be applied to those other events or if it is specific to this type of nested event. Additionally, it may be worth commenting on what would lead to the absence or presence of a nested structure.

2) I think some further discussion of the origin of the additional radially inward electric field that is not balanced by the diamagnetic field. Does the presence of this radially inward electric field imply the presence of a divergence (or in this case a convergence) of the electric field and thus a small charge separation? How does this mesh with the quasi-neutrality assumption? If the currents are completely supported by the electron diamagnetic drift the Hall and electron pressure terms will completely cancel out (in the perpendicular direction), so I think the reason the Hall term is stronger and there is an additional electric field is a relevant question, which hasn't been fully explained in the current manuscript.

3) I think the manuscript could use some further discussion of why such magnetic cavities are significant in a turbulent plasma. Do these structures have a dynamical significance within the turbulence or are they simply a curiosity that plays a passive role in the dynamics? What new information is gained from this new model and will it help to address these questions?

Minor comments:

Line 43: I think "environment" should be "environments".

Line 56: "Alfven" should have an accent mark.

Line 128 and 154: What is meant by plasma rest frame? Is this taken to be the local ion flow frame, some sort of average flow frame?

Line 149: Is this non-gyrotropic feature associated with a non-gyrotropic distribution (in the electron flow frame) or is it just a signature of the offset of the distribution associated with the electron fluid flow?

Figs. 1 and 3: Is there a significance to the difference between the observed and model E_y on all of the spacecraft? In the observations it seems to consistently have a bipolar signature, while in the model there is a unipolar signature.

Response to Reviewers

First of all, we are grateful to all the three reviewers for their constructive comments, which certainly provide us the opportunity to improve our paper. We have carefully considered these comments and accordingly revised the manuscript. Please find below a point-to-point response to these comments, in which the reviewers' comments are shown in blue and our replies are in black.

To Reviewer #1

The submitted paper is a continuation of the paper published in NCOMMS by Liu et al. (2019), see ref. 7, where some members of this team have analyzed MMS data on kinetic scales of only tens of kilometers. Using the unprecedented very high-resolution of 30 (150) milliseconds for electron (proton) distribution functions, and 8 milliseconds for magnetic field instruments, they have found that the magnetic hole is nearly circular with a radius of about 10 km, i.e. extremely small in the space environment. Now the present authors compare MMS observations with the results of a selfconsistent kinetic theory. The paper is also interesting and worth publishing for a general audience, but I am concerned about the following issues.

Abstract

the title: magnetic cavities or better magnetic holes (see lines 45, 46)

We agree that the term 'magnetic hole' has been used more often than 'magnetic cavity' in previous studies, even if they refer to essentially the same kind of structures. However, in the Liu et al. (2019) paper, the term 'magnetic cavities' was used (as requested by the referee). Since our paper is a continuation of Liu et al. (2019), we believe that it is most appropriate to use 'magnetic cavities' as well, and clearly state in the very first paragraph that magnetic cavities are 'also referred to as magnetic holes' (see line 54 in the revised manuscript).

line: 35 (and 84 – 86) It is not clear from the abstract why it is important to predict measurements from one to a second/third MMS spacecraft. Maybe because of an anomaly on MMS 4 from June/July 2018, e.g., see and add ref. in the text *Astrophys. J. Lett*, 885:L26 (2019).

We believe that our model, or any scientific hypothesis, must provide testable predictions so that its validity can be examined from the observational/experimental data. In our paper, the testable predictions are the virtual observations obtained from the model (shown in the right panels of Figure 3), in which no MMS3 or MMS4 data were used to determine the model parameters. The direct comparison between these testable predictions and the observations from MMS3 and MMS4 spacecraft (shown in the left panels of Figure 3), together with their excellent consistency, give us the confidence on the model validity. It is also the foundation

of our further analysis (such as confirming the quasi-equilibrium state of magnetic cavities and resolving the cavity profiles based on single-spacecraft measurements).

The event analyzed in our paper occurred in 2015, and the magnetic cavities were observed by MMS1, MMS3 and MMS4. Therefore, it is probably not associated with the MMS4 anomaly in 2018.

Model

Please refer to all panels in Fig. 1 in the text (not only in captions) possibly in order.

Thank you for your advices. We have adjusted the panel orders in Figure 1, with the electron bulk velocity and the electric field observations moved towards the bottom panels, to have the order consistent with the order in the text.

line 113: Why MMS2 observed no perturbations?

For this event, the radius of the electron-scale cavity was approximately 10 km (see Figure 4), whereas MMS2 was 15 km away from MMS1 when the latter intersected the cavity center. In other words, MMS2 was simply outside the electron-scale magnetic cavity. In the revised manuscript, we have stated in lines 132-134 that the MMS interspacecraft separation was by the order of 10 km, and in line 139 that the absence of magnetic perturbations in MMS2 indicates the very small size of the magnetic cavity.

Line 120 ff: e.g. discuss the depressed magnetic field in Fig. 1a

These are the most characteristic signatures of magnetic cavities. In the revised manuscript, we have provided further details on the magnetic field depression level, that is, from 32 nT to 24 nT in the proton scale and from 24 nT to 17 nT in the electron scale (see lines 146 and 150).

line 142 ff: What are the other interpretations?

The alternative interpretation is actually one of the main indications of our model, given in detail in lines 291-307. In brief, the perpendicular and parallel anisotropy inside and outside the electron-scale cavity (most clearly shown in Fig. 1g and 1j) corresponds to the positive and negative μ -dependence of phase space densities for current-carrying and background electron populations, respectively.

I. 146 the apparent non-gyrotropic features has been reported also from other spacecraft, e.g., THEMIS, ApJL, 851:L42, 2017.

In the revised manuscript, we describe in further detail the nature of the non-gyrotropic electron distributions (see lines 222-233). The electron non-gyrotropy stems partially from the dependence of electron phase space density on canonical angular momentum, and

partially from the coexistence of two electron components with different angular bulk velocities. The resultant non-Gaussian distributions are also manifested by the difference between pressure components $P_{e,\rho\rho}$ and $P_{e,\varphi\varphi}$ (see Figure 4g and lines 240-241). We have also cited the Egedel et al. (2016) and the Macek et al. (2017) papers in lines 242-244 for similar non-gyrotropic distributions in the reconnection region and in the turbulent magnetosheath, respectively.

I. 183: show proton and electron gyroscapes on the axes and explain and justify the speculation that magnetic depressions are caused by diamagnetic currents.

Thank you for the great suggestion. In the revised manuscript, the horizontal axis provides the radial distance in three different units (kilometers, thermal proton gyroradii, and thermal electron gyroradii).

To show the relationship between magnetic depressions and the diamagnetic motion, we have added a new panel (panel b) in Figure 4 showing the current densities associated with the cross-scale magnetic depressions. This panel can be compared directly with the proton and electron bulk velocities given in Figures 4i and 4j. One can clearly see that outside the electron-scale cavity, the current is carried solely by the proton diamagnetic motion (the red line in Figure 4i). Within the electron-scale cavity, the electron diamagnetic motion (the red line in Figure 4j) provides ~70% of the current density, with the remaining ~30% contributed by the electron $E \times B$ motion. More detailed descriptions on the current carriers are given in lines 256-268 in the revised manuscript.

The discrepancy on the basis of your kinetic model (line 192ff):

line 202: why only diagonal terms are taken in Eq. (1). In the reconnection region (line 207, see e.g., ref. ApJL 885:L26, 2019) the electrons are decoupled from ions are not anymore convected by the magnetic field, and the equilibrium fluid equation is not anymore applicable. In the generalized Ohm's law of Eq. (4) an inertial term is missing. It should be justify why this term is not important.

In our model, the nondiagonal terms of the pressure tensor $\mathbf{P} = m \int (\mathbf{v} - \mathbf{u})(\mathbf{v} - \mathbf{u}) f d^3\mathbf{v}$ is zero because of the symmetric distributions of f in the radial and in the z directions. In the plasma multi-fluid theory, the momentum equation of the electron fluid is derived by simply taking the 1st-order moment of the Vlasov equation over the velocity space (the detailed derivation can be found in Baumjohann and Treumann, *Basic Space Plasma Physics*, equations (7.6)-(7.11)). Therefore, we can still use the electron momentum equations even if the electrons and ions are decoupled (as long as the Vlasov equation remains valid).

Regarding the generalized Ohm's law, the reviewer is correct that we neglect the inertial term, which in our modeled equilibrium corresponds to the centrifugal force of the azimuthal electron flows. The reason is that according to our calculations, the centrifugal force is about two orders of magnitude smaller than either the Lorentz force or the pressure divergence force. But we agree with the reviewer that we should have made it more accurate. In the

revised manuscript, we have added the inertial term into equations (1) and (4), and clearly stated that the centrifugal force is much smaller than the other forces so that we can neglect the inertial term in both equations (see lines 236-238 and 274-276)

I. 244: At the electron scale, when electrons decouple, the divergence term dominates the electric field and in the case of reconnection it should be even stronger than the Hall (opposite to your case as you say in the parenthesis, see e.g., GRL 46, 10295; ApJL 885:L26, 2019).

We agree that both the Hall term and the pressure divergence term should be important in any small-scale structures including the electron diffusion region of magnetic reconnection, and their relative importance may depend case by case. For example, the Hall term was stronger than the pressure divergence term in Torbert et al. (2016), whereas the pressure divergence term became dominant in Macek et al. (2019a; 2019b). In the revised manuscript, we cited all these papers to indicate the similarities between the magnetic cavities and reconnection diffusion regions, and the complexities of these electron-scale structures (see lines 286-290).

In summary: please say more clearly about any physical interpretation justifying that your equilibrium model agrees with observations. Why a theoretical solution is 'hard to derive'? Discuss possible applications for multi-scales observations in space plasmas.

The agreement between our equilibrium model and the observations indicates that the electron-scale magnetic cavities can survive over a macro time scale (rather than being non-steady magnetic field variations). Therefore, they are able to transport the hot trapped electrons away from the source, and shape the spectrum of compressional fluctuations on very small scales.

The statement that theoretical solutions are 'hard to derive' was made in Haynes et al. (2015), which justified their motivation of using particle-in-cell simulations (rather than theoretical solutions) to understand the magnetic cavity formation. In this paper, we provide the first theoretical solution that matches the observational data, and we believe that this solution can be used for analysis of plasma waves and instabilities (that have been widely observed within magnetic cavities) and/or serve as the initial condition for kinetic simulations. In the revised manuscript, we have largely rewritten the introduction (lines 43-78, 99-105) and the summary (lines 327-334) sections, to highlight the motivations and significance of our study.

To Reviewer #2

The paper presents observations from the MMS spacecraft of electron and ion scale magnetic cavities to which several dynamical features are associated (e.g., density enhancement, local electron perpendicular heating). The observations are compared to

synthetic data obtained using a virtual spacecraft that crosses similar cavities obtained from a theoretical model built by the authors. A very good agreement is found between the model-based data and those of MMS. This type of approach allows one to gain insight into the dynamics of magnetic cavities in space plasmas. The paper is concise and well written, although a bit too technical.

Ion and electron scales magnetic cavities have been previously reported from MMS observations, some of the related papers are cited by the authors. The added value of the present paper is the theoretical model that provides a possible framework to interpret MMS observations of the magnetic cavities. This is an interesting piece of work that helps better understanding the dynamics of this type of coherent structures forming turbulent plasmas.

This is a very good summary of what we have done in this paper, and we appreciate the reviewer for acknowledging the validity of our model

However, the paper appears to this referee as incremental and of a low-to-moderate impact to the current knowledge in the field. As said above, the observations are not new and other theoretical/numerical works have been already proposed (although not necessarily with the same level of detail). It is true that the evidenced structures can locally energize particles, but what is the impact of such energization on the dynamics on larger (macro) scales or on the modelling of general problems of plasma physics is not clear. This contrasts with the claim in the abstract about the potential broad impact of the work (e.g., "Understanding the nature of cross-scale structures ubiquitous as magnetic cavities is important to assess the energy partition, coupling and cascade in the plasma universe"). Also, the multi-scale coupling claimed in different locations of the paper is not strongly supported by the observations: the sole co-existence of two coherent structures with two characteristic scales –proton and electron gyro-radii, can hardly be considered as a multi-scale coupling as known in turbulent or complex systems.

For all these reason I do not recommend the publication of the paper in Nature Com. I would rather recommend submitting it to a more specialized journal in space or plasma physics (e.g., JGR, JGR, Phys. Plasmas) where similar papers are regularly published (after considering the comments above).

The reviewer is right that there are other kinetic models of magnetic cavities, especially the Shustov et al. (2016) equilibrium model. However, the Shustov et al. (2016) model could not reproduce an important observational signature, the anisotropic and nongyrotropic electron distributions in Figures 1h-1k, which cast a serious doubt on whether the observed magnetic cavities can be described by equilibrium models at all. If not, can the electron-scale cavities be stable and quasi-stationary, or are we simply observing some non-steady magnetic field variations?

This question becomes more important in the context of compressional and incompressional fluctuations and their roles in plasma turbulence. It is usually believed that incompressional

waves are damped very slowly due to the lack of Landau resonance, which enables efficient energy transport and therefore largely shapes the turbulent electromagnetic field spectra. On the other hand, the compressional fluctuations are believed to contribute to a very small energy fraction (up to 10%) in plasma turbulence. Although previous reports on magnetic cavities have suggested that the compressional waves may still survive over a long time (if their growth is sufficiently strong to reach a nonlinear stage), we are not sure if this is still the case for electron-scale magnetic cavities since the energy dissipation in the electron kinetic scales could be strong enough to decay the turbulent spectrum (Alexandrova et al., 2009). The excellent agreement between our equilibrium model and the observations provides a concrete answer to these questions. We are now more confident that the cross-scale cavities can travel with the plasma flows for a macro time scale, to transport the trapped electron population away from the energy release source and shape the spectrum of compressional fluctuations on very small scales. In the revised manuscript, we have largely rewritten the introduction and the summary sections to highlight the motivations and significance of our study.

We also agree with your comments that the coexistence of electron- and proton-scale cavities does not necessarily mean the occurrence of strong cross-scale coupling processes. Although it is possible that the parallel electron anisotropy in the center of the proton-scale cavity may provide an environment favorable for sporadic commencement of the electron mirror instability and consequently enable formation of electron-scale cavities (Haynes et al., 2015), we don't have enough evidence to show that this is exactly what happened. We have accordingly removed most claims on multi-scale coupling processes, except that we keep the discussions in lines 324-327 on the Haynes et al. (2015) hypothesis mentioned above.

To Reviewer #3

This manuscript by Li et al. constructs a self-consistent, cylindrically symmetric, equilibrium model for the electromagnetic fields and particle distribution function within nested electron and ion scale magnetic cavities, by considering the conservation of energy, canonical angular momentum, and magnetic moment. The model is applied to observations of nested ion and electron-scale magnetic cavities by the Magnetospheric Multiscale mission, which have been discussed from an observational perspective. The model appears to successfully reproduce many features observed within the observations, not only for the one spacecraft that was used to set the parameters of the model, but also for the other two spacecraft which encountered the structure.

Kinetic scale magnetic cavities have been reported across a number of different space plasma environments, including the solar wind, Earth's magnetosheath, Earth's magnetopause, and Earth's plasma sheet, and more recently nested ion and electron-scale cavities have been reported. Such structures have been suggested to be related to turbulence in these regions and may play some role dissipation, although, to my knowledge,

a full understanding of their formation and significance within the plasma has not been fully determined. This paper is significant in that it presents a new self-consistent model for the fields and particle distribution functions.

Overall, I find the paper well written and believe it will likely be suitable for publication with some revision and further discussion. In particular, I think further discussion of the significance and usefulness of the new model would be warranted, particularly for a Nature publication, and I have some questions about the origin of the electric fields in the structure that may prompt further discussion within the paper. I have outlined this and other comments in detail below.

1) Does this model only work for “electron-vortex” magnetic cavities that are embedded within ion-scale cavities? To my knowledge, most of the observed electron-scale magnetic cavities are not reported within a larger ion scale cavity, so I think it is important to discuss whether this model could be applied to those other events or if it is specific to this type of nested event. Additionally, it may be worth commenting on what would lead to the absence or presence of a nested structure.

This is a very good question, and the short answer is yes. The model can be used to describe single-scale (either electron- or ion-scale) magnetic cavities as well. Below are two examples of the modeled single-scale cavities.

The left panel shows the magnetic field variations within a proton-scale cavity, with the horizontal axis presenting the distance from the cavity center in logarithmic scale. The model parameters are: $\delta = 0.88$, $N_e = 23.7 \text{ cm}^{-3}$, $\Omega_e = 0.073 \text{ s}^{-1}$, $\Omega_i = -0.711 \text{ s}^{-1}$, $\theta_{e,0} = 52 \text{ eV}$, $\theta_{e,1} = 78 \text{ eV}$, $\theta_{i,0} = 265 \text{ eV}$, $\theta_{i,1} = 473.3 \text{ eV}$, $b_{e,0} = 0 \text{ nT}$, and $b_{e,1} = 0 \text{ nT}$. The right panel shows an electron-scale cavity, with the following parameters: $\delta = 0.88$, $N_e = 23.7 \text{ cm}^{-3}$, $\Omega_e = 451 \text{ s}^{-1}$, $\Omega_i = 0 \text{ s}^{-1}$, $\theta_{e,0} = 52 \text{ eV}$, $\theta_{e,1} = 78 \text{ eV}$, $\theta_{i,0} = 265 \text{ eV}$, $\theta_{i,1} = 473.3 \text{ eV}$, $b_{e,0} = 0 \text{ nT}$, and $b_{e,1} = 0 \text{ nT}$.

We point out here that the cavity scale is largely determined by the $\theta_{\alpha,1}/\Omega_\alpha$ values, with the subscript α representing the proton and electron species. Therefore, similar values between $\theta_{i,1}/\Omega_i$ and $\theta_{e,1}/\Omega_e$ would degenerate the multi-scale model into a single-scale cavity. This relationship can be qualitatively understood by the fact that the cavity scale depends on two factors, the characteristic current density and the magnetic field difference across the cavity. Here the first factor is associated with the ion and electron angular bulk velocity Ω_α , and the

second factor (essentially the thermal pressure difference, since the magnetic and thermal pressures are balanced) is determined by the temperature $\theta_{\alpha,1}$. Similar relationships can be also found in the classical Harris current sheet (Harris, 1962). Since these arguments may be too technical for the paper, we have decided to add only a few sentences to briefly discuss this effect (see lines 366-369 in the revised manuscript).

2) I think some further discussion of the origin of the additional radially inward electric field that is not balanced by the diamagnetic field. Does the presence of this radially inward electric field imply the presence of a divergence (or in this case a convergence) of the electric field and thus a small charge separation? How does this mesh with the quasi-neutrality assumption? If the currents are completely supported by the electron diamagnetic drift the Hall and electron pressure terms will completely cancel out (in the perpendicular direction), so I think the reason the Hall term is stronger and there is an additional electric field is a relevant question, which hasn't been fully explained in the current manuscript.

The reviewer is right that the nonzero electric field divergence implies a minor charge separation. However, if we calculate the net charge density from the electric field divergence, we could find that it is about 7 orders of magnitude smaller than the number density of protons or electrons. In other words, the Poisson's equation can be safely approximated by the quasi-neutrality condition. This approximation is described in Francis Chen's classical textbook (*Introduction to Plasma Physics and Controlled Fusion*, Volume 1, Chapter 3.6) as the plasma approximation, which applies in most cases with spatial scales larger than the Debye length. It is advised in Chen's textbook that the electric field should be derived from the momentum equation rather than Poisson's equation (unless high-frequency motions are involved in which the electron inertia becomes important). This also explains why the Hall and the electron pressure terms do not cancel out in the perpendicular direction. We have revised the relevant descriptions in lines 382-386.

3) I think the manuscript could use some further discussion of why such magnetic cavities are significant in a turbulent plasma. Do these structures have a dynamical significance within the turbulence or are they simply a curiosity that plays a passive role in the dynamics? What new information is gained from this new model and will it help to address these questions?

These are very important questions. In brief, the new information from our model (and its agreement with the observations) is that the cross-scale cavities can travel with the plasma flows for a macro time scale, and they can shape the spectrum of compressional fluctuations on very small scales. We have largely rewritten the introduction section (especially the first three paragraphs) and the summary section (the last paragraph), to highlight the motivation and significance of our study.

Minor comments:

Line 43: I think "environment" should be "environments".

Corrected. Thank you.

Line 56: "Alfven" should have an accent mark.

Nice catch! Corrected.

Line 128 and 154: What is meant by plasma rest frame? Is this taken to be the local ion flow frame, some sort of average flow frame?

By 'plasma rest frame', we meant the rest frame of the proton fluid. We agree that this is not very accurate, and in the revised manuscript, the term 'plasma rest frame' has been replaced by 'proton fluid rest frame'.

Line 149: Is this non-gyrotropic feature associated with a non-gyrotropic distribution (in the electron flow frame) or is it just a signature of the offset of the distribution associated with the electron fluid flow?

This is a very good question, and the short answer is that even in the electron fluid rest frame the nongyrotropic distribution still exists. The figure below shows the modeled electron distribution function in the electron fluid rest frame, from which the electron nongyrotropy can be clearly seen.

The reasons are two-fold. First of all, the distributions of the current-carrying electrons (equation (10) in the Methods section) are not a simple Maxwellian or shifted-Maxwellian due to their dependence on both the canonical angular momentum and the magnetic moment. Moreover, the coexistence of two electron components with different bulk velocities would contribute to the nongyrotropic distributions (even if each component were gyrotropic). In the revised manuscript, we have described this effect in lines 222-233.

Figs. 1 and 3: Is there a significance to the difference between the observed and model E_y on all of the spacecraft? In the observations it seems to consistently have a bipolar signature, while in the model there is a unipolar signature.

Thank you for drawing our attention to this minor difference. The E_y field observed by MMS1 indicates the existence of azimuthal electric field, which is not included in our model. This electric field is in the same direction as the electric current, which indicates the conversion of electromagnetic energy to plasma thermal and/or kinetic energy. A possible mechanism to generate the azimuthal electric field is electromagnetic induction associated with cavity shrinkage, a process reported recently via MMS observations (Liu et al., 2019, GRL). This could indeed happen in this event, since the observed magnetic field (in Fig. 1e) variations are slightly asymmetric with sharper gradient in the trailing edge.

To make sure that the cavity shrinkage can produce bipolar E_y variations, we develop an *ad hoc* model based on the Faraday's law of induction. The four panels of the figure below, from top to bottom, present the spatial profiles of magnetic field B_z (at two different times), the B_z time derivative, the azimuthal electric field (determined from Faraday's law), and the corresponding E_y field along MMS1 trajectories, respectively. One can clearly see from this simple model the bipolar E_y variations, which are quite similar to the MMS1 observations in Figure 1m.

These features, however, are beyond the scope of our equilibrium model. Therefore, we have only briefly discussed this potential process in the revised manuscript (lines 308-319). In the future, we may carry out a simulation study to analyze the particle kinetics associated with cavity shrinkage, in which our equilibrium model could provide an initial condition.

REVIEWER COMMENTS

Reviewer #1 (Remarks to the Author):

In my view, the authors have taken into consideration my suggestions and the manuscript is now improved. Because the results are new and important for turbulence in plasma I recommend the paper for publication in NatComm.

Minors:

ref. 52, capitalize THEMIS

ref. 54, comma missing 46,

Reviewer #2 (Remarks to the Author):

Review of "Nested, electron- and proton-scale magnetic cavities: Manifestation of kinetic theta-pinch equilibrium in space plasmas"

Li et al.

In the revised paper the authors considered my comments. As I said in my previous report, the manuscript presents interesting theoretical and observational results related to nonlinear structures in magnetized space plasmas and their potential role in energizing the plasma particles. My major criticism, which still holds, does not concern the physics itself but rather the limited impact of the reported results, which led me to conclude that the paper is more suitable for publications in space of plasma physics journals. I let the Editorial board make a decision on this point. Below, I provide my last comments/corrections to further improve the paper, regardless of where it would eventually be published.

1. Lines 61-64: the statement in this new paragraph is not correct. The observation of a decaying power spectrum (regardless of its scaling) does not contradict (or prevent) the existence of small (electron) scale structures. This is clearly shown for instance in Perri et al., PRL, 2012 based on the same data analyzed in Sahraoui et al. PRL 2009 that showed the first electron dissipation range. This

is also known in hydrodynamics turbulence where the dissipation range can be filled by stretching vorticity structures. A similar observation can be made as well for ion scale range: the spectra can decay rapidly (over a small scale range) while it still can be populated by different structures such as vortices and current sheets.

Note also that the physics of electron scales is still not fully understood: its scaling is still debated (Sahraoui et al., 2013) and the existence of an “ultimate” electron cascade range is more than plausible according to many theoretical and numerical prediction (e.g., Scheckochihin et al., 2009; Camporeale & Burgess, ApJ, 2011). I ask the authors to rephrase the corresponding sentences by rather emphasizing the possible interplay between the observed electron scale structures and turbulence cascade to control the dynamics at those scales, and to include the additional references that I provide here.

2. I agree with the authors on the potential role of compressible fluctuations in plasma dynamics in the magnetosheath and even in the solar wind (although they are weaker in the latter). However, many references that discussed explicitly this point are missing, see for instance Hadid et al., 2017 (solar wind) and Hadid et al., PRL 2018 (magnetosheath) and the references therein.

3. Minor:

- a. Line 74: “ ... existence of stable magnetic ...”
- b. Line 74: “ ... AS to which mechanism dominates ...”
- c. Line 75: “... may not be easy TO ACHIEVE in the electron”
- d. Line 97: “superimposed” instead of “superposed”

Reviewer #3 (Remarks to the Author):

The authors have addressed many of my concerns from the previous round of reviews. I find this to be a very interesting paper and overall I find that it is worth publication. It provides a detailed, self-consistent, kinetic model of magnetic cavities (holes) within plasmas, which allows the detailed examination of the electric fields and particle distributions that support these structures. The model will likely provide a useful tool for exploring the role of these structures within turbulent plasmas and minor differences between the model and the observed structures may provide the basis for future study. My only remaining comment is a continuation of one of my previous comments about the origin of the addition radial electric field from the last round of reviews, which I have outlined below.

- With regards to the additional radial electric field associated with the Hall term, which goes beyond the electron diamagnetic field, I agree with all of the authors points in response to my previous comment. However, I think it may also be worth highlighting in the manuscript the role of the ions in supporting this Hall electric field in the electron cavity. While, as the authors point out on lines 256-262, the ion velocity is zero within the electron-scale cavity, I believe the fact that this is achieved by the ExB drift being balanced by an ion diamagnetic drift means that the ion pressure gradient is playing a role in supporting the current that generates the Hall electric field in the electron scale cavity. While this may be a somewhat subtle point, I think the fact that a significant portion of the discussion is dedicated to using the model to understand the origin of the electric fields and currents within the cavities, makes this a relevant point to mention.

Response to Reviewers

First of all, we are grateful to all the three reviewers for their constructive comments. We have carefully considered these comments and accordingly revised the manuscript. Please find below a point-to-point response to these comments, in which the reviewers' comments are shown in blue and our replies are in black.

To Reviewer #1

In my view, the authors have taken into consideration my suggestions and the manuscript is now improved. Because the results are new and important for turbulence in plasma, I recommend the paper for publication in NatComm.

Minors:

ref. 52, capitalize THEMIS

ref. 54, comma missing 46

Both corrected. Thank you!

To Reviewer #2

In the revised paper the authors considered my comments. As I said in my previous report, the manuscript presents interesting theoretical and observational results related to nonlinear structures in magnetized space plasmas and their potential role in energizing the plasma particles. My major criticism, which still holds, does not concern the physics itself but rather the limited impact of the reported results, which led me to conclude that the paper is more suitable for publications in space of plasma physics journals. I let the Editorial board make a decision on this point. Below, I provide my last comments/corrections to further improve the paper, regardless of where it would eventually be published.

1. Lines 61-64: the statement in this new paragraph is not correct. The observation of a decaying power spectrum (regardless of its scaling) does not contradict (or prevent) the existence of small (electron) scale structures. This is clearly shown for instance in Perri et al., PRL, 2012 based on the same data analyzed in Sahraoui et al. PRL 2009 that showed the first electron dissipation range. This is also known in hydrodynamics turbulence where the dissipation range can be filled by stretching vorticity structures. A similar observation can be made as well for ion scale range: the spectra can decay rapidly (over a small scale range) while it still can be populated by different structures such as vortices and current sheets.

Thank you for the correction. We have removed the statement, and accordingly revised the sentence on the discovery of electron-scale magnetic cavities and their embedding within

proton-scale cavities. See lines 60-65 in the revised manuscript.

Note also that the physics of electron scales is still not fully understood: its scaling is still debated (Sahraoui et al., 2013) and the existence of an “ultimate” electron cascade range is more than plausible according to many theoretical and numerical prediction (e.g., Scheckochihin et al., 2009; Camporeale & Burgess, ApJ, 2011). I ask the authors to rephrase the corresponding sentences by rather emphasizing the possible interplay between the observed electron scale structures and turbulence cascade to control the dynamics at those scales, and to include the additional references that I provide here.

We have rephrased the sentence and added a few more sentences to emphasize the possible relationship between the observed magnetic cavities and the turbulent cascade in the electron scale. Also, the references provided (Sahraoui et al., 2013; Scheckochihin et al., 2009; Camporeale & Burgess, 2011), together with a few relevant references (Galtier et al., 2000; Sahraoui et al., 2009; Andres et al., 2019), have been included in the revised manuscript. See lines 48-50, 52, 334-340.

2. I agree with the authors on the potential role of compressible fluctuations in plasma dynamics in the magnetosheath and even in the solar wind (although they are weaker in the latter). However, many references that discussed explicitly this point are missing, see for instance Hadid et al., 2017 (solar wind) and Hadid et al., PRL 2018 (magnetosheath) and the references therein.

Thank you for bringing our attention to these papers. The Hadid et al. papers have now been cited in the revised manuscript.

3. Minor:

a. Line 74: “ ... existence of stable magnetic ...”

b. Line 74: “ ... AS to which mechanism dominates ...”

c. Line 75: “... may not be easy TO ACHIEVE in the electron ...”

d. Line 97: “superimposed” instead of “superposed”

Thank you very much for pointing them out. They are all corrected.

To Reviewer #3

The authors have addressed many of my concerns from the previous round of reviews. I find this to be a very interesting paper and overall I find that it is worth publication. It provides a detailed, self-consistent, kinetic model of magnetic cavities (holes) within plasmas, which allows the detailed examination of the electric fields and particle distributions that support these structures. The model will likely provide a useful tool for exploring the role of these structures within turbulent plasmas and minor differences between the model and the

observed structures may provide the basis for future study. My only remaining comment is a continuation of one of my previous comments about the origin of the additional radial electric field from the last round of reviews, which I have outlined below.

With regards to the additional radial electric field associated with the Hall term, which goes beyond the electron diamagnetic field, I agree with all of the authors points in response to my previous comment. However, I think it may also be worth highlighting in the manuscript the role of the ions in supporting this Hall electric field in the electron cavity. While, as the authors point out on lines 256-262, the ion velocity is zero within the electron-scale cavity, I believe the fact that this is achieved by the $E \times B$ drift being balanced by an ion diamagnetic drift means that the ion pressure gradient is playing a role in supporting the current that generates the Hall electric field in the electron scale cavity. While this may be a somewhat subtle point, I think the fact that a significant portion of the discussion is dedicated to using the model to understand the origin of the electric fields and currents within the cavities, makes this a relevant point to mention.

We fully agree, and we are grateful to the reviewer for these comments. We have added a few sentences (see lines 265-270 and line 286) to highlight the point that the current density profile can be determined by the sum of electron and ion pressure divergences even if the ion bulk flow is negligible within the electron-scale cavity.

REVIEWERS' COMMENTS

Reviewer #3 (Remarks to the Author):

Overall, I find that the authors have reasonably addressed my previous comments, as well as those of the other reviewers and that the manuscript is suitable for publication in the present form. As I was specifically asked to review the comments and responses of reviewer 2 as well, I will also note that, while I personally find this manuscript to be a particularly interesting theoretical study, I can to some extent see where reviewer 2 is coming from in their critique on the suitability of the manuscript for publication in Nature Communications specifically. As reviewer 2 notes, this is to some extent an editorial decision. From my point of view, within a field directly related to the topic, while the study certainly isn't the final word on magnetic holes in collisionless plasmas, it does develop a new theoretical tool that I think will be particularly useful both for the future analysis of magnetic holes in observations, as well as, potentially, for application in numerical studies examining the evolution of such structures and their impact on the plasma and on turbulent dynamics. I, therefore, would recommend it for publication in Nature Communications.

I do have one final, rather minor, comment:

Line 335: The authors use the phrase "ultimate cascade region" based on reviewer 2's comment. I personally find this to be a little bit cryptic in the manuscript and would suggest clarifying a bit. I guess what is meant by this phrase is that the dynamics in the electron scales could be influenced by a nonlinear cascade, as opposed to just being a "dissipation range" for the turbulence.

Response to Reviewer

We are grateful to the reviewer for his/her support to our study. We have also incorporated the minor comments into our paper (in which the revised words are highlighted in red). Please find below a point-to-point response to the comments, in which the reviewer's comments are shown in blue and our replies are in black.

To Reviewer #3

Overall, I find that the authors have reasonably addressed my previous comments, as well as those of the other reviewers and that the manuscript is suitable for publication in the present form. As I was specifically asked to review the comments and responses of reviewer 2 as well, I will also note that, while I personally find this manuscript to be a particularly interesting theoretical study, I can to some extent see where reviewer 2 is coming from in their critique on the suitability of the manuscript for publication in Nature Communications specifically. As reviewer 2 notes, this is to some extent an editorial decision. From my point of view, within a field directly related to the topic, while the study certainly isn't the final word on magnetic holes in collisionless plasmas, it does develop a new theoretical tool that I think will be particularly useful both for the future analysis of magnetic holes in observations, as well as, potentially, for application in numerical studies examining the evolution of such structures and their impact on the plasma and on turbulent dynamics. I, therefore, would recommend it for publication in Nature Communications.

Once again, thank you very much!

I do have one final, rather minor, comment: Line 335: The authors use the phrase "ultimate cascade region" based on reviewer 2's comment. I personally find this to be a little bit cryptic in the manuscript and would suggest clarifying a bit. I guess what is meant by this phrase is that the dynamics in the electron scales could be influenced by a nonlinear cascade, as opposed to just being a "dissipation range" for the turbulence.

We agree that the phrase 'ultimate cascade region' could be confusing. In the revised manuscript, we avoid this phrase and change the corresponding sentence into the followings: 'Given the complicated dynamics of energy cascade and dissipation^{63,64} indicated by the presence of distinct breakpoints in the turbulent magnetic spectrum at electron gyroscscales^{65,66}, the analyzed magnetic cavities may also be considered intermittent structures produced by the inhomogeneity of electron-scale turbulence cascade.'